# On Producing Shortest Cost-Optimal Plans

**Michael Katz[1], Gabriele Röger[2], Malte Helmert[2]**

[1]IBM Research, Yorktown Heights, NY, USA
[2]University of Basel, Basel, Switzerland
michael.katz1@ibm.com, {gabriele.roeger,malte.helmert}@unibas.ch

## Abstract

Cost-optimal planning is at the heart of planning research, with many existing planners that produce provably optimal solutions. While some applications pose additional restrictions, such as producing shortest (in the number of actions) among the cost-optimal plans, standard cost-optimal planning does not provide such a guarantee. We discuss two possible approaches to produce provably the shortest among the cost-optimal plans, one corresponding to an instantiation of cost-algebraic A$^*$, the other based on a cost transformation. We formally prove that the new cost-transformation method indeed produces the shortest among the cost-optimal plans and empirically compare the performance of the approaches in different configurations.

## 1 Introduction

The aim of classical planning is to determine a plan, which is a sequence of deterministic actions that leads from a given initial state to a goal state. Each action is associated with a non-negative cost and the cost of a plan is defined as the sum of these action costs. In the special case of *optimal* planning, only the plans with *minimal cost among all plans* are considered valid solutions.

Alternative optimization criteria have been considered in the literature: in multi-objective search (Stewart and White 1991; Mandow and la Cruz 2005) there are multiple – typically incomparable and conflicting – cost functions and the aim is to identify all plans with non-dominated cost vectors. Other recent planning variants where the solution consists of entire sets of plans are diverse planning (Nguyen et al. 2012) with the aim to identify a set of qualitatively different plans, top-k planning (Katz et al. 2018) with the aim to identify a set of plans such that no better plan outside the set exists, and top-quality planning (Katz, Sohrabi, and Udrea 2020) with the aim to identify all plans of some bounded quality.

In this paper, we consider the problem of finding a cost-optimal plan that has a minimal number of actions among all cost-optimal plans, or shortest cost-optimal plan. This is for example required as a subroutine for top-quality planning (Katz, Sohrabi, and Udrea 2020; Katz and Sohrabi 2022) but also an interesting property if the plan gets presented to humans, as it is for example the case in human-in-the-loop planning or explainable AI planning where the optimality is

measured with respect to the observer model (Chakraborti et al. 2021).

Note that the problem of finding a shortest cost-optimal plan is different from the multi-objective case. We indeed have two cost measures here, namely plan cost and plan length, but these are not pari passu. If there is a plan $\pi$ of length 1 (i.e. a single action application) and cost 5 but there is another plan of cost 4 then $\pi$ is not considered a valid solution, even if the other plan consists of hundreds of actions. So we do not give up the requirement of cost-optimal plans, we only have the additional requirement that among these we want a shortest one in the number of actions.

This concept of shortest cost-optimal plans is loosely related to the concept of strongly optimal plans in stubborn set pruning (Wehrle and Helmert 2014). These are optimal plans with a minimal number of *zero-cost* actions. The difference is that we consider the number of *all* actions, independent of their cost. Fišer, Torralba, and Shleyfman (2019) on the other hand, define the concept of strongly optimal plans to be optimal plans with a minimal number of actions. To avoid confusion, we use the notion of shortest cost-optimal plans.

Edelkamp, Jabbar, and Lluch Lafuente (2005) already considered the problem of finding optimal paths relative to cost notions that can be very different from the standard sum of action costs. They introduce so-called *cost algebras*, which are a very general concept covering a wide range of cost measures, including the number of actions as well as the sum of action costs. Moreover, both these cost measures are *strictly isotone*, a property that Edelkamp, Jabbar, and Lluch Lafuente identified to be sufficient for their prioritized Cartesian product being again a cost algebra. This prioritized Cartesian product corresponds exactly to the case we consider in this paper: Prefer cheaper plans, and among equally cheap plans, prefer the shorter one.

The paper is structured as follows: We first formally introduce the problem and present the cost-algebraic A$^*$ algorithm in the specific instantiation for our cost functions. We then present a cost transformation that allows to solve the problem with standard A$^*$ and discuss different ways to set parameters for the approach. For integer costs, we show that a specific configuration of cost-algebraic A$^*$ is equivalent to a certain configuration of the cost transformation approach. Finally, we empirically evaluate the different approaches.

## 2 Shortest Cost-Optimal Planning Problem

We consider tasks that are given by their transition system $\Pi = \langle S, A, T, cost, s_0, S_* \rangle$, where $S$ is a finite set of *states*, $s_0 \in S$ is the *initial* state and $S_* \subseteq S$ is the set of *goal states*. Set $A$ is the finite set of *actions*, where the *action cost function cost* : $A \to \mathbb{R}^{0+}$ assigns every action a non-negative cost. The *transition relation* $T \subseteq S \times A \times S$ is deterministic, i.e. for every state $s$ and action $a$, there is at most one $s'$ with $(s, a, s') \in T$. If there is such an $s'$, we say that $a$ is *applicable* in $s$ and that $s'$ is the successor state of applying $a$ in $s$. An *s-plan* is a sequence of actions that is consecutively applicable in state $s$ and where the final state is a goal state. A *plan* is an $s_0$-plan. We will sometimes focus on *simple* applicable action sequences, i.e. applicable action sequences that traverse no state more than once.

For defining the cost of an action sequence, we extend the cost function from single actions to action sequences as $cost(\langle a_1, \ldots, a_n \rangle) := \sum_{i=1,\ldots,n} cost(a_i)$. With $|\pi|$ we denote the *length* of action sequence $\pi$, i.e. $|\langle a_1, \ldots, a_n \rangle| = n$.

We denote by $\leq_\Pi$ the partial order of action sequences defined by their cost, i.e., $\pi \leq_\Pi \pi'$ iff $cost(\pi) \leq cost(\pi')$. By $\preceq_\Pi$ we denote the partial order defined by their costs and length, i.e., $\pi \preceq_\Pi \pi'$ if $cost(\pi) < cost(\pi')$ or if $cost(\pi) = cost(\pi')$ and $|\pi| \leq |\pi'|$.

In standard cost-optimal planning we must for a given task $\Pi$ identify a plan that is minimal among all plans with respect to $\leq_\Pi$ or detect that the task is unsolvable, i.e. it has no plan. In contrast, in the *shortest cost-optimal planning problem* we must find a plan for $\Pi$ that is minimal among all plans with respect to $\preceq_\Pi$ or again detect $\Pi$ as unsolvable.

## 3 Cost-Algebraic Approach

As already mentioned in the introduction, our shortest cost-optimal planning problem is a special case of the problems covered by cost-algebraic heuristic search (Edelkamp, Jabbar, and Lluch Lafuente 2005).

We will now briefly describe the specific instantiation of the cost-algebraic A* algorithm for our scenario.

First, we need a heuristic for the cost-algebraic problem. Such a cost-algebraic heuristic maps every state to a pair $\langle h_c, h_d \rangle \in \mathbb{R}^{0+} \times \mathbb{R}^{0+}$. The heuristic is *admissible* if for every state $s$ it holds that $h_c \leq c^*$ and $h_d \leq d^*$, where $c^*$ is the cost of a cheapest plan for state $s$ and $d^*$ is the length of a shortest among these cheapest plans.

The standard A* algorithm (Hart, Nilsson, and Raphael 1968) maintains an open list of search nodes, where each search node is associated with a state and with an implicit path from the initial state to this state. In each iteration, A* pops a node from the open list, which in the standard scalar case is ordered by the $f$-values of the nodes. Here, $f = g_c + h_c$, where $g_c$ is the cost of the associated path from the initial node to the associated state and $h_c$ is the heuristic estimate for this state. If the popped node is a goal node, the search terminates. Otherwise it expands the node by generating all successor nodes and adding the expanded state to the closed list. Each generated node that is not associated with a closed state is added to the open list. Also nodes with closed states are added to open, if the associated path to the state is

cheaper than the previous one. This is called reopening and only necessary if the heuristic does not have the additional property of being consistent.

The cost-algebraic A* variant extends standard A* in two ways: the open list prioritizes by $\langle f_c, f_d \rangle$, where $f_c = g_c + h_c$ and $f_d = g_d + h_d$ with cost-algebraic heuristic estimate $\langle h_c, h_d \rangle$, $g_c$ being the cost of the associated path and $g_d$ being the length of the associated path. So $f_c$ corresponds to the $f$-value in the standard case and $f_d$ is the analogous value in terms of plan length. Pair $\langle f_c, f_d \rangle$ has a higher priority than $\langle f_c', f_d' \rangle$ if $f_c < f_c'$ or if $f_c = f_c'$ and $f_d < f_d'$.

The second extension is the reopening criterion. A state is not only reopened if the search encounters a cheaper path to the state but also if it encounters an equally cheap but shorter path to the state.

The results by Edelkamp, Jabbar, and Lluch Lafuente imply that with these two changes, the algorithm is guaranteed to produce shortest cost-optimal plans.

To determine an admissible cost-algebraic heuristic in practice, we can use any (standard) admissible heuristic to determine the cost estimate $h_c$. For the distance estimate $h_d$, it is not necessary to only consider minimum-cost plans but also a lower-bound on the length of *any* plan will be an admissible estimate (but possibly lower than necessary). For this reason, we can simply use a standard admissible heuristic, but evaluate it with respect to a different cost function that assigns every action a cost of 1.

Of course the computation of an additional distance estimate causes computational overhead and it is not immediately obvious that it will pay off: since the first component $f_c$ dominates the ordering of the open list, the A* search will have to consider all action sequences where $f_c$ is strictly lower than the cost of an cost-optimal plan. Hence, the distance information can only be beneficial for the ordering on the last $f_c$ layer. For this reason, we will also consider the extreme case where we set the second heuristic component $h_d$ to constant 0. This is still an admissible estimate, so also this much cheaper configuration guarantees shortest cost-optimal plans.

## 4 Cost Transformation Approach

The second approach that we consider will use standard cost-optimal solvers but ensures shortest cost-optimal plans by means of a transformation of the action cost function. Exchanging the cost function does not alter the set of applicable action sequences but will only influence what plans are considered optimal. The idea of the transformation is to increase the cost of every action by a small $\epsilon$, so that a plan with $n$ actions will accumulate an additional cost of $n \cdot \epsilon$. We will choose $\epsilon$ small enough so that for any relevant action sequences $\pi$ and $\pi'$, if $\pi$ has originally been strictly cheaper than $\pi'$, this will stay true also with the additionally incurred cost.

**Theorem 1** *For every planning task $\Pi$ there is a task $\Pi'$ that only differs from $\Pi$ in the action cost function, such that for all applicable simple action sequences $\pi$ and $\pi'$ it holds that $\pi \preceq_\Pi \pi'$ iff $\pi \leq_{\Pi'} \pi'$.*

**Proof:** Consider planning task $\Pi = \langle S, A, T, cost, s_0, S_* \rangle$. Let $P$ be the set of all applicable simple action sequences in $\Pi$. Since the sequences are simple, the length of any $\pi \in P$ is bounded by the number of states $|S|$. Since moreover the set of actions $A$ is finite, $P$ is a finite set. Let $\delta > 0$ be smaller or equal to the smallest *non-zero* cost difference $|cost(\pi) - cost(\pi')|$ between any two sequences $\pi, \pi' \in P$ with $cost(\pi) \neq cost(\pi')$. Let $L$ be larger than the length difference $||\pi| - |\pi'||$ of any two sequences in $\pi, \pi' \in P$.

We define $\epsilon := \frac{\delta}{L} > 0$ and a new cost function $cost_{+\epsilon}$ as $cost_{+\epsilon}(a) = cost(a) + \epsilon$. Let $\Pi'$ be defined as $\Pi$ only with cost function $cost_{+\epsilon}$, i.e. $\Pi' = \langle S, A, T, cost_{+\epsilon}, s_0, S_* \rangle$.

Let $\pi, \pi' \in P$. We distinguish two cases.

If $cost(\pi) = cost(\pi')$, then $\pi \preceq_\Pi \pi'$ iff $|\pi| \leq |\pi'|$ iff $cost_{+\epsilon}(\pi) = cost(\pi) + \epsilon|\pi| \leq cost(\pi') + \epsilon|\pi'| = cost_{+\epsilon}(\pi')$ iff $\pi \leq_{\Pi'} \pi'$.

For the case $cost(\pi) \neq cost(\pi')$, we consider the two directions separately:

If $\pi \preceq_\Pi \pi'$ and $cost(\pi) \neq cost(\pi')$ then $cost(\pi) < cost(\pi')$. Since the cost difference $cost(\pi') - cost(\pi)$ is non-zero, it must be $\geq \delta$. By the choice of $L$, $|\pi| - |\pi'| < L$. We get that $cost_{+\epsilon}(\pi') - cost_{+\epsilon}(\pi) = cost(\pi') + \epsilon|\pi'| - cost(\pi) - \epsilon|\pi| = cost(\pi') - cost(\pi) - \epsilon(|\pi| - |\pi'|) > cost(\pi') - cost(\pi) - \epsilon L = cost(\pi') - cost(\pi) - \delta \geq 0$. Overall, this implies $cost_{+\epsilon}(\pi') > cost_{+\epsilon}(\pi)$, thus $\pi \leq_{\Pi'} \pi'$.

If $\pi \leq_{\Pi'} \pi'$ then $cost_{+\epsilon}(\pi) \leq cost_{+\epsilon}(\pi')$. By the definition of $cost_{+\epsilon}$, we have $cost(\pi) + \epsilon|\pi| \leq cost(\pi') + \epsilon|\pi'|$ (*). Since $|\pi'| - |\pi| < L$, it holds that $\epsilon(|\pi'| - |\pi|) < \epsilon L = \delta$. With (*), we get $\delta > \epsilon(|\pi'| - |\pi|) \geq cost(\pi) - cost(\pi')$. By the definition of $\delta$, $\delta > cost(\pi) - cost(\pi')$ implies that $cost(\pi) \not\succ cost(\pi')$. Since we currently consider the case where $cost(\pi) \neq cost(\pi')$, this implies $cost(\pi) < cost(\pi')$ and thus $\pi \preceq_\Pi \pi'$. $\square$

Based on Theorem 1, one can find shortest cost-optimal plans by finding cost-optimal plans for the transformed cost task. The restriction to simple action sequences is not a limitation for this purpose: shortest cost-optimal plans for the original task are always simple because otherwise there were shorter plans of equal or lower cost (cutting out the cycle); cost-optimal plans in the transformed task are simple because there are no 0-cost actions in this task (which could be used to form a cycle without increasing the cost).

The proof of the theorem is somewhat unsatisfactory, because we only establish the *existence* of a suitable value $\epsilon$. As the length of any simple applicable action sequence is between 0 and $|S| - 1$, we can use the trivial bound of $|S|$ for $L$. For the common case of integer action costs, we can set $\delta$ to 1, which gives $\epsilon = 1/|S|$. But since $|S|$ is typically very high, $\epsilon$ would be very small. This is problematic because planners often only support integer costs, so we need to scale up the cost function $cost_{+\epsilon}$ to integers, which can lead to extremely high action costs and overflow problems in the planner. For this reason, we are interested in a cost function that serves the same purpose with lower action costs.

We actually do not need the full generality of Theorem 1 because we do not need to cover *all* applicable simple action sequences but only those relevant to the search. For this reason, we first revisit the proof of Theorem 1 for a task *with*

*integer costs* to identify a sufficient requirements for $\epsilon$ and each two applicable simple action sequences that we compare during the search:

- If $cost(\pi) = cost(\pi')$ then any positive $\epsilon$ is sufficient.
- For $\pi \preceq_\Pi \pi'$ implying $\pi \leq_{\Pi'} \pi'$ if $cost(\pi) \neq cost(\pi')$, we only need that $\epsilon(|\pi| - |\pi'|) < \delta$. With integer costs and $\delta = 1$, it is thus sufficient for the proof if the length difference of the compared action sequences is strictly smaller than $1/\epsilon$.
- For $\pi \leq_{\Pi'} \pi'$ implying $\pi \preceq_\Pi \pi'$ if $cost(\pi) \neq cost(\pi')$, we only exploit from $\epsilon$ that $\epsilon(|\pi'| - |\pi|) < \delta$. Again it is sufficient if the length difference of the compared action sequences is strictly smaller than $1/\epsilon$.

So overall, it is sufficient if $\epsilon > 0$ and the length difference of any two compared action sequences is smaller than $1/\epsilon$. Intuitively, this makes sense because the additional action cost we incur when accounting for the length of the sequence will never outweigh the original cost of any action application, at least on the action sequences we consider.

What action sequences must cost-algebraic A* consider? Obviously, those where the search node would actually be popped from the open list in cost-algebraic A* are sufficient. For standard scalar A*, it is well known that it expands all nodes that have a lower $f$-value than the optimal solution cost $C^*$, and a subset of the nodes, where the $f$-value equals $C^*$ (last $f$-layer; depending on tie-breaking) but no nodes with higher $f$-value. Since in cost-algebraic A* the $f_c$-value for the cost dominates the ordering of the open list, it also must expand all nodes with $f_c < C^*$ but from the nodes on the last $f_c$-layer (with $f_c = C^*$), it will only consider a subset of those, where $f_d$ is less than or equal to the length of a shortest optimal plan. We do not have much information about the heuristics for cost and distance, but we know that their estimates are non-negative. So we can conclude that cost-algebraic A* will never pop an action sequence $\pi$ from the open list for which $cost(\pi) > C^*$.

Let us now combine these two lines of thought and let $M$ be some integer that is strictly larger than the length of any applicable simple action sequence $\pi$ with $cost(\pi) \leq C^*$.

From a theoretical perspective, we can always use $|S|$ for $M$ but this more precise definition allows us to exploit additional insights we might have into the problem at hand. With our previous consideration, any $\epsilon > 0$ with $M \leq 1/\epsilon$ will work in the transformation (still considering tasks with integer costs). In particular, we can set $\epsilon$ to $1/M$. With this choice, we get $cost_{+\epsilon}(a) = cost(a) + 1/M$ and $M cost_{+\epsilon}(a) = M cost(a) + 1$. Since all action costs with $cost_{+\epsilon}$ are positive, scaling them with constant factor $M$ does not affect the ordering of action sequences by their cost. Overall, we observe that with integer action costs this function $M cost_{+\epsilon}$ (henceforth referred to as $cost_M$) is a cost function suitable for the cost-transformation-based approach to finding shortest optimal plans. Moreover, it can also be used with planners that only support integer action costs, because it maps only into the integers.

In what follows, we will frequently refer to this cost function and the cost transformed task. Therefore, we include it in the following definition.

**Definition 1** *For task $\Pi = \langle S, A, T, cost, s_0, S_* \rangle$ and $M \in \mathbb{N}_{>0}$, we define cost function $cost_M$ as $cost_M(a) = M \cdot cost(a) + 1$ and task $\Pi_M$ as $\Pi_M = \langle S, A, T, cost_M, s_0, S_* \rangle$.*

So far our discussion was driven by the behaviour of cost-algebraic A$^*$ and indeed this will be relevant for Theorem 3 that establishes that in a certain configuration both approaches will behave equally. If we now focus only on the resulting scalar problem and the aim to identify shortest optimal plans, can we decrease $M$ below the previous criterion?

Consider applicable action sequences $\pi$ of length $|\pi| \geq M$ and $\pi'$ with $cost(\pi') > cost(\pi)$ and length $|\pi'| < |\pi| - M(cost(\pi') - cost(\pi))$. If both sequences are actual plans, $\pi$ would be better wrt. the optimization criterion because of its lower cost. But under cost $cost_M$, we have $\pi' \leq_{\Pi_M} \pi$, so A$^*$ potentially would not consider $\pi$ but terminate with $\pi'$. Is this always a problem or put differently, are there applicable action sequences a search could skip while still finding an optimal solution? Obviously, all action sequences that cannot be extended to an optimal plan can be pruned from the search space without harming completeness or the optimality guarantee. Due to the monotonicity of the plan length, this is the case for all action sequences that are longer than the length of a shortest optimal plan.

Hence, if we are only interested into finding a shortest optimal plan for task $\pi$ and we know that the solution will be shorter than some $M \in \mathbb{N}$ then we can simply solve the induced problem $\Pi_M$ with standard A$^*$.

## Heuristics

If we use a heuristic search planner to solve the cost-transformed task, we have some freedom what cost function we use for the heuristic computation. The first and most obvious option we will consider is to compute the heuristic estimates with respect to the transformed cost function.

However, with this cost function there can be a huge number of action sequences that all exhibit different costs. This can slow down the heuristic computation. For this reason, we will also consider the alternative approach of computing heuristic estimates with the original cost function and scaling them by $M$. These estimates are admissible with respect to the transformed cost:

**Theorem 2** *Consider task $\Pi$ with cost function $cost$ and a $M \in \mathbb{N}_{>0}$. If $h$ is an admissible estimate for state $s$ in $\Pi$ then $Mh$ is an admissible estimate for $s$ in $\Pi_M$.*

**Proof:** Let $\pi$ be a plan for state $s$ that is optimal with respect to cost function $cost_M$ and let $h$ be an admissible estimate for $s$ wrt. to cost function $cost$. Since $\pi$ is an $s$-plan, it holds that $cost(\pi) \geq h$. As $cost_M(\pi) = Mcost(\pi) + |\pi| \geq Mcost(\pi) \geq Mh$, value $Mh$ is an admissible estimate for $s$ wrt. cost function $cost_M$. $\qquad \square$

Interestingly, in the typical setup with integer action costs and integer heuristic estimates, A$^*$ with this second heuristic approach leads to the same search behaviour as cost-algebraic A$^*$ using the same heuristic for cost and the blind estimator that always sets $h_d$ to 0 for distance.

**Theorem 3** *Consider task $\Pi = \langle S, A, T, cost, s_0, S_* \rangle$ with $cost : A \to \mathbb{N}_0$, heuristic $h_c : S \to \mathbb{N}_0$ that is admissible wrt. cost and heuristic $h_d$ with $h_d(s) = 0$ for all $s \in S$. Let $l$ be the length of the longest action sequence associated with a search node expanded by cost-algebraic A$^*$ on $\Pi$ with the given heuristics.*

*For integer $M > l$, standard A$^*$ on $\Pi_M$ with heuristic $Mh_c$ will expand nodes in the same order as cost-algebraic A$^*$ on $\Pi$ with heuristics $h_c$ and $h_d$ if the algorithms use the same-tie breaking.*

**Proof:** We show the theorem by induction over the expansions of the two A$^*$ variants. Let *Open* and *Closed* be the open and closed list of the standard A$^*$ configuration and *Open*$_{ca}$ and *Closed*$_{ca}$ be the lists of the cost-algebraic A$^*$ configuration. Our induction hypothesis is that after $k$ expansions, *Open* will contain the same nodes as *Open*$_{ca}$ and *Closed* will contain the same nodes as *Closed*$_{ca}$. Moreover, if a node $n$ has priority $\langle p_c, p_d \rangle$ in *Open*$_{ca}$ then node $n$ has priority $Mp_c + p_d$ in *Open*.

After 0 expansions, both closed lists are empty and both open lists contain the node for the initial state, associated with the empty action sequence. In the cost-algebraic configuration, it has priority $\langle h_c(s_0), 0 \rangle$ and in the scalar configuration, it has priority $Mh_c(s_0)$.

For the inductive step, we first show that both versions can (and with equal tie-breaking will) expand the same node.

If $n$ is a node with minimum priority $\langle p_c, p_d \rangle$ in *Open*$_{ca}$, then all other nodes $n'$ have priority $\langle p'_c, p'_d \rangle$ with (1) $p'_c > p_c$ or with (2) $p'_c = p_c$ and $p'_d \geq p_d$. For all nodes $n'$ with (2), $Mp_c + p_d \leq Mp'_c + p'_d$ trivially holds, so *Open* does not prefer $n'$ over $n$. For all nodes $n'$ with (1) we know that $p'_c \geq p_c + 1$ because the priorities are integers. Hence, $Mp'_c \geq Mp_c + M$. Note that $p_d$ and $p'_d$ correspond to the lengths of the associated paths. By the requirements on $l$ and $M$, this implies that $p_d - p'_d < l < M$, so $Mp'_c > Mp_c + p_d - p'_d$ or, equivalently, $Mp'_c + p'_d > Mp_c + p_d$. These are the priorities of $n'$ and $n$ in *Open*, so $n$ must be preferred over $n'$.

Let $n$ be a node with minimum priority in *Open*, let $\pi$ be the associated path and $s$ be the reached state. For all other nodes $n'$ (with $\pi'$ and $s'$ being defined analogously), it holds that $Mcost(\pi) + |\pi| + Mh_c(s) \leq Mcost(\pi') + |\pi'| + Mh_c(s')$ (*). The corresponding priorities of the nodes in *Open*$_{ca}$ are $\langle cost(\pi) + h_c(s), |\pi| \rangle$ and $\langle cost(\pi') + h_c(s'), |\pi'| \rangle$, respectively. If $cost(\pi) + h_c(s) < cost(\pi') + h_c(s')$, the *Open*$_{ca}$ must trivially prefer $n$ over $n'$. If $cost(\pi) + h_c(s) = cost(\pi') + h_c(s')$ then (*) implies that $|\pi| \leq |\pi'|$, so $n'$ again does not get preferred over $n$. The case $cost(\pi) + h_c(s) > cost(\pi') + h_c(s')$ is impossible: costs and heuristic estimates are integers, so $cost(\pi) + h_c(s) + 1 \geq cost(\pi') + h_c(s')$. Multiplying by $M$ and combining with (*) gives $|\pi'| - |\pi| \geq M$ but by the requirements on $l$ and $M$, $|\pi'| - |\pi| < M$ (or no node with a sequence of length $|\pi'|$ gets expanded, also preventing $n'$).

So both configurations select the same node $n$ for expansion and subsequent addition to the closed list. Since the cost functions do not affect the set of successor nodes but only the associated costs, both variants will consider the same successors. Let $\pi$ be the action sequence associated with $n$

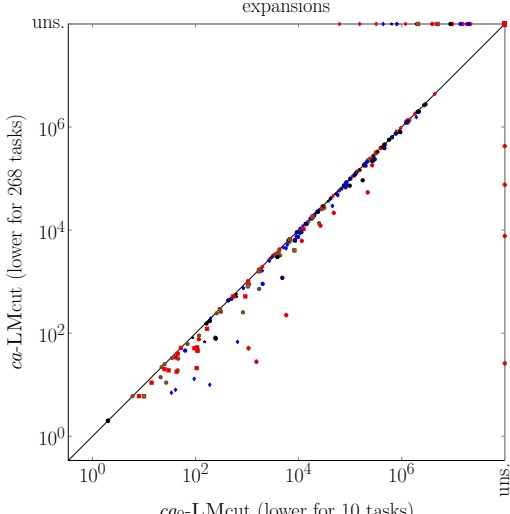
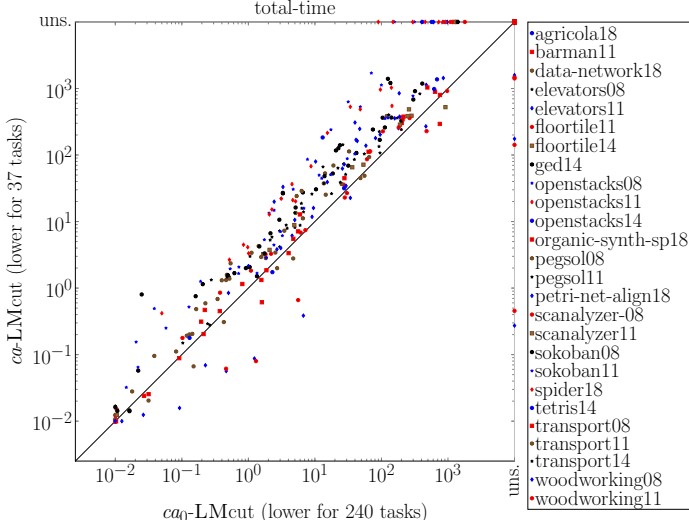

Figure 1: Number of expanded nodes (left) and total time (right), comparing the cost-algebraic approach ($ca$) with informed distance estimates to the ($ca_0$) approach using constant 0.

and consider the successor with action $a$, leading to state $s$. If the corresponding successor node $n'$ is not yet in the closed list (which contains the same nodes in both configuration), both configurations add it to their open list. In the cost-algebraic configuration, $n'$ has priority $\langle cost(\pi) + cost(a) + h_c(s), |\pi| + 1 \rangle$. The same node in the scalar configuration has priority $Mcost(\pi) + |\pi| + Mcost(a) + 1 + Mh_c(s) = M(cost(\pi) + cost(a) + h_c(s)) + (|\pi| + 1)$, thus maintaining the connection from the induction hypothesis. In the following we refer to the action sequence for $n'$ by $\pi'$ (= $\pi$ extended with $a$).

Consider the case that $s$ was already included in the closed lists, where at the last expansion of a corresponding node it was associates with some action sequence $\pi''$. This addition to closed has happened after the same number of expansions with the same action sequence $\pi''$ in both configurations. We need to show that either both of the configurations add $n'$ to their open list (re-opening the state) or none of them.

The scalar configuration adds $n'$ to *Open* iff $Mcost(\pi') + |\pi'| < Mcost(\pi'') + |\pi''|$ (**). The cost-algebraic configuration adds $n'$ to *Open*$_{ca}$ iff $cost(\pi') < cost(\pi'')$ or if $cost(\pi') = cost(\pi'')$ and $|\pi'| < |\pi''|$. We now show that the two conditions are equivalent.

($\Rightarrow$) From (**), we get that $M(cost(\pi') - cost(\pi'')) < |\pi''| - |\pi'|$. Since $|\pi'| \leq l$, $|\pi''| \leq l$ and $M > l$, it holds that $|\pi''| - |\pi'| < M$. Together, we get $cost(\pi') - cost(\pi'') < 1$, so $cost(\pi') < cost(\pi'') + 1$. As all values are integers, this implies $cost(\pi') \leq cost(\pi'')$. If $cost(\pi') = cost(\pi'')$ then (**) directly implies $|\pi'| < |\pi''|$.

($\Leftarrow$) If $cost(\pi') = cost(\pi'')$ and $|\pi'| < |\pi''|$, we get (**) directly. If $cost(\pi') < cost(\pi'')$, then $cost(\pi') + 1 \leq cost(\pi'')$ because costs are integers. Thus $Mcost(\pi') + M \leq Mcost(\pi'')$ and since (by the analogous reasoning as in the other direction) $|\pi'| - |\pi''| < M$, we get (**). $\square$

## 5 Experimental Evaluation

To empirically compare the various suggested methods, we have implemented these methods on top of the Fast Downward planning system (Helmert 2006). The code is available at https://github.com/ibm/shortest-optimal-downward. The experiments were performed on Intel(R) Xeon(R) Gold 6248 CPU @2.50GHz machines, with the time and memory limit of 30min and 3.5GB, respectively. The benchmark set consists of all STRIPS benchmarks with non-unit costs (with the exception of parcprinter[1] ) from optimal tracks of International Planning Competitions 1998-2018, a total of 587 tasks in 28 domains. We exclude unit-cost tasks because for these all optimal plans are shortest optimal plans. As all tasks have integer costs and Fast Downward supports integer costs only, in order to obtain a cost transformation into integer values, we assume a bound of $M = 10000$ on the plan length. Transformed action costs $cost_M(a)$ are therefore $10000 \cdot cost(a) + 1$. As there are no existing planners that produce shortest optimal plans, the suggested methods are compared to each other, with a variety of admissible heuristics. The heuristics we experiment with are the blind heuristic, LMcut (Helmert and Domshlak 2009), merge-and-shrink abstraction (denoted by M&S) (Helmert, Haslum, and Hoffmann 2007), counterexample-guided Cartesian abstraction refinement (denoted by CEGAR) (Seipp and Helmert 2018), $h_{max}$ (Bonet and Geffner 2001), and pattern database heuristic iPDB (Haslum et al. 2007). We measure the search progress in terms of the number of expanded nodes as well as the number of heuristic evaluations; for the required time, we distinguish search

---

[1]The parcprinter domain has actions with costs in hundreds of thousands. After cost transformation the domain would have actions with costs in billions.

| Coverage | LMcut | | M&S | | CEGAR | | $h_{max}$ | | iPDB | |
|---|---|---|---|---|---|---|---|---|---|---|
| | $ca_0$ | $ca$ | $ca_0$ | $ca$ | $ca_0$ | $ca$ | $ca_0$ | $ca$ | $ca_0$ | $ca$ |
| data-network18 (20) | 12 | 12 | 9 | 9 | 14 | 14 | 11 | 11 | **12** | 11 |
| elevators08 (30) | **22** | 19 | 15 | 15 | 20 | 20 | 17 | 17 | 23 | 23 |
| elevators11 (20) | **18** | 17 | 12 | 12 | 17 | 17 | **14** | 13 | 18 | 18 |
| floortile11 (20) | **7** | 6 | 4 | 4 | 2 | 2 | 6 | 6 | 2 | 2 |
| floortile14 (20) | **6** | 5 | 2 | 2 | 0 | 0 | 5 | 5 | 0 | 0 |
| openstacks08 (30) | **22** | 16 | 22 | 22 | **22** | 20 | **22** | 21 | 22 | 22 |
| openstacks11 (20) | **17** | 10 | 17 | 17 | **17** | 15 | **17** | 16 | 17 | 17 |
| openstacks14 (20) | **3** | 1 | 3 | 3 | 3 | 3 | 3 | 3 | 3 | 3 |
| organic-s-sp18 (20) | 14 | 14 | **7** | 6 | 9 | 9 | **19** | 18 | 7 | 6 |
| pegsol08 (30) | 27 | 27 | **29** | 28 | 28 | 28 | 27 | 27 | 29 | 29 |
| pegsol11 (20) | 17 | 17 | **19** | 18 | 18 | 18 | 17 | 17 | 19 | 19 |
| petri-net-align18 (20) | **9** | 8 | 4 | 4 | 1 | 1 | **11** | 9 | 0 | 0 |
| scanalyzer-08 (30) | **11** | 9 | 12 | 12 | 12 | 12 | 9 | 9 | 13 | 13 |
| scanalyzer11 (20) | **7** | 6 | 9 | 9 | 9 | 9 | 6 | 6 | 10 | 10 |
| sokoban08 (30) | **29** | 28 | 27 | 27 | 23 | 23 | 28 | 28 | **30** | 29 |
| sokoban11 (20) | 20 | 20 | 20 | 20 | 20 | 20 | 20 | 20 | **20** | 19 |
| spider18 (20) | **11** | 8 | 13 | 13 | **11** | 10 | 9 | 8 | **13** | 3 |
| tetris14 (17) | 5 | 5 | 7 | 7 | **9** | 8 | **9** | 7 | 1 | 1 |
| transport11 (20) | 6 | 6 | 6 | 6 | 6 | 6 | 6 | 6 | **12** | 11 |
| transport14 (20) | 6 | 6 | 7 | 7 | 7 | 7 | **8** | 6 | 9 | 9 |
| woodworking08 (30) | 13 | **17** | 14 | 14 | 10 | **11** | 10 | 10 | 12 | 12 |
| woodworking11 (20) | 8 | **12** | 9 | 9 | 5 | **6** | 5 | 5 | 7 | 7 |
| **Sum other (90)** | 30 | 30 | 34 | 34 | 30 | 30 | 34 | 34 | 37 | 37 |
| **Sum (587)** | **320** | 299 | **301** | 298 | **293** | 289 | **313** | 302 | **316** | 301 |

Table 1: Per-domain coverage for the cost algebra methods.

time (used by the A*algorithms) and total time (including heuristic pre-computations); coverage refers to the number of problems solved.

## 5.1 Cost-Algebraic Approach

As already discussed in Section 3, using a distance estimator in cost-algebraic A* can only be beneficial for the ordering of the last $f_c$ layer, so it is worth evaluating whether the additional effort for a distance estimator still pays off.

For this reason, we compare the cost-algebraic approach ($ca$) with informative heuristic functions for both, cost and distance estimates, to the one that only uses an informed heuristic for cost estimates but constant 0 for the distance estimate ($ca_0$). For simplicity, we use the same heuristic for both cost and distance estimates, simply replacing all action costs with 1 for the distance estimate. Figure 1 (left) compares $ca$-LMcut to $ca_0$-LMcut in terms of expanded nodes: when using an additional heuristic for distance estimation the number of expanded nodes indeed decreases. However, as per-node evaluation time also increases for most tasks this does not pay off in terms of total time (Figure 1, right). This translates directly to the coverage of the two approaches: $ca$-LMcut solves 299 tasks overall, compared to 320 of $ca_0$-LMcut. The only domains where $ca$-LMcut has a better coverage are the woodworking domains (17 vs. 13 and 12

vs. 8). Table 1 shows per-domain coverage results for multiple heuristics ("other" aggregates domains where $ca$ and $ca_0$ perform equally). The best results per domain and heuristic are bolded. The results are consistent across the various heuristics. Notably, the largest difference in coverage is on the spider domain, 13 for $ca_0$-iPDB vs. 3 for $ca$-iPDB.

Overall, the reduction in the search effort from using an informed distance estimator does not to outweigh its computation cost, both time- and coverage-wise, for all tested heuristics. For this reason, in the following experiments we will only compare to the better-performing variant $ca_0$.

## 5.2 Cost-Transformation Approach

For the cost-transformation-based approach, there is only a single heuristic for estimating the cost to go, but no separate distance estimator. One obvious option for the heuristic is to compute it directly from the cost-transformed task (= the standard behaviour of Fast Downward). We denote this configuration by $ct$. The other option is to use the cost-transformed task for the computation of $g$-values in the search but to compute the heuristic estimates on the original task and scaling them with constant factor $M$. The resulting estimates are admissible for the cost-transformed task by Theorem 2. We denote this second approach by $wct$.

Table 2 shows the per-domain coverage comparison. Observe that, with the exception of $h_{max}$ heuristic, these two methods are quite complementary across the tested heuristics, and the difference in coverage on a particular domain can be quite large.

While the heuristic computed directly on the cost-transformed task ($ct$) is more informed, the transformed cost function can have a negative impact on the computation-time of some heuristics. We observe that the overhead incurred by the cost transformation is very different for the different heuristics. But also if we fix the heuristic, it depends on the domain whether the impact is positive or negative. Both effects are very pronounced in the LMcut heuristic, hence we present more details for this heuristic. Figure 2 shows the details of the comparison in terms of the number of heuristic evaluations per second[2], as well as the number of expanded nodes. We see the dominance of the heuristic computation used in $ct$ clearly reflected in the number of expansions (right). But if we have a look at the time required for the heuristic computation (left), we observe that the computation on the original cost function with subsequent scaling is almost consistently faster. The speedup is most prominent on the openstacks domains, which also benefit most from the $wct$ configuration in terms of coverage.

Overall, the direct computation is better-informed but can have a negative impact on the computation time with some heuristics. Whether the potentially lower computation time of the scaling approach outweighs the somewhat lower heuristic guidance depends on the balance of the two effects (speedup and impact on guidance), which varies from domain to domain.

---

[2]Tasks with search time under 1 second or the number of evaluation is under 100 were not plotted.

| Coverage | blind | | LMcut | | M&S | | CEGAR | | $h_{max}$ | | iPDB | |
|---|---|---|---|---|---|---|---|---|---|---|---|---|
| | ct | wct | ct | wct | ct | wct | ct | wct | ct | wct | ct | wct |
| data-network18 (20) | 7 | 7 | 12 | 12 | 9 | 9 | 13 | **14** | 11 | 11 | 12 | 12 |
| elevators08 (30) | 14 | 14 | 21 | **22** | 15 | **16** | **21** | 20 | 17 | 17 | **25** | 23 |
| elevators11 (20) | 12 | 12 | 16 | **18** | 13 | 13 | **18** | 17 | 14 | 14 | **19** | 18 |
| floortile11 (20) | 2 | 2 | 7 | 7 | 4 | 4 | 2 | 2 | 6 | 6 | **3** | 2 |
| openstacks08 (30) | 22 | 22 | 15 | **22** | 22 | 22 | **25** | 22 | 22 | 22 | 16 | **22** |
| openstacks11 (20) | 17 | 17 | 10 | **17** | 17 | 17 | **19** | 17 | 17 | 17 | 11 | **17** |
| openstacks14 (20) | 3 | 3 | 1 | **3** | 3 | 3 | **4** | 3 | 3 | 3 | 2 | **3** |
| organic-s-sp18 (20) | 10 | 10 | 13 | **14** | 6 | 6 | 9 | 9 | 19 | 19 | 6 | **7** |
| pegsol08 (30) | 27 | 27 | 27 | 27 | 29 | 29 | 28 | 28 | 28 | 28 | **30** | 29 |
| pegsol11 (20) | 17 | 17 | 17 | 17 | 19 | 19 | 18 | 18 | 18 | 18 | **20** | 19 |
| petri-net-align18 (20) | 4 | 4 | 8 | **9** | 0 | **4** | 1 | **2** | 10 | **11** | 0 | 0 |
| scanalyzer-08 (30) | 12 | 12 | **15** | 10 | **13** | 12 | 12 | 12 | 9 | 9 | 13 | 13 |
| scanalyzer11 (20) | 9 | 9 | **12** | 7 | **10** | 9 | 9 | 9 | 6 | 6 | 10 | 10 |
| sokoban08 (30) | 22 | 22 | 29 | 29 | 27 | 27 | **24** | 23 | 28 | 28 | 30 | 30 |
| spider18 (20) | 11 | 11 | 10 | **11** | 13 | 13 | 11 | 11 | 9 | 9 | 3 | **13** |
| woodworking08 (30) | 8 | 8 | **17** | 13 | **15** | 14 | **11** | 10 | 11 | 11 | **13** | 12 |
| woodworking11 (20) | 3 | 3 | **12** | 8 | **10** | 9 | **6** | 5 | 6 | 6 | **8** | 7 |
| **Sum other (187)** | 71 | 71 | 73 | 73 | 76 | 76 | 72 | 72 | 82 | 82 | 79 | 79 |
| **Sum (587)** | 271 | 271 | 315 | **319** | 301 | **302** | **303** | 294 | 316 | **317** | 300 | **316** |

Table 2: Per-domain coverage for the cost transformation methods.

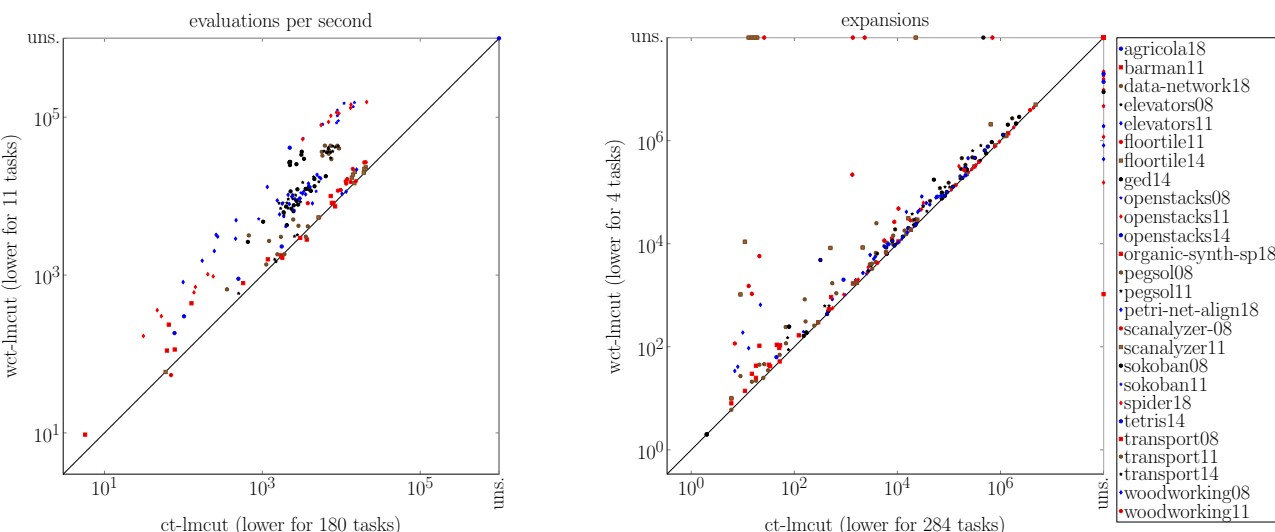

Figure 2: Number of heuristic evaluations per second (left) and expansions (right), comparing the both cost-transformation configurations *ct* and *wct* with the LMcut heuristic.

## 5.3 Cost-Algebraic vs. Cost-Transformation

Theorem 3 guarantees that the search behavior of $ca_0$ and *wct* to be exactly the same, that is, they examine the same nodes in the same order. However, the computational overhead can still be different. For this reason, we also experimentally compare these two configurations to each other.

We indeed can empirically verify that both methods always result in the exact same number of expansions. Figure

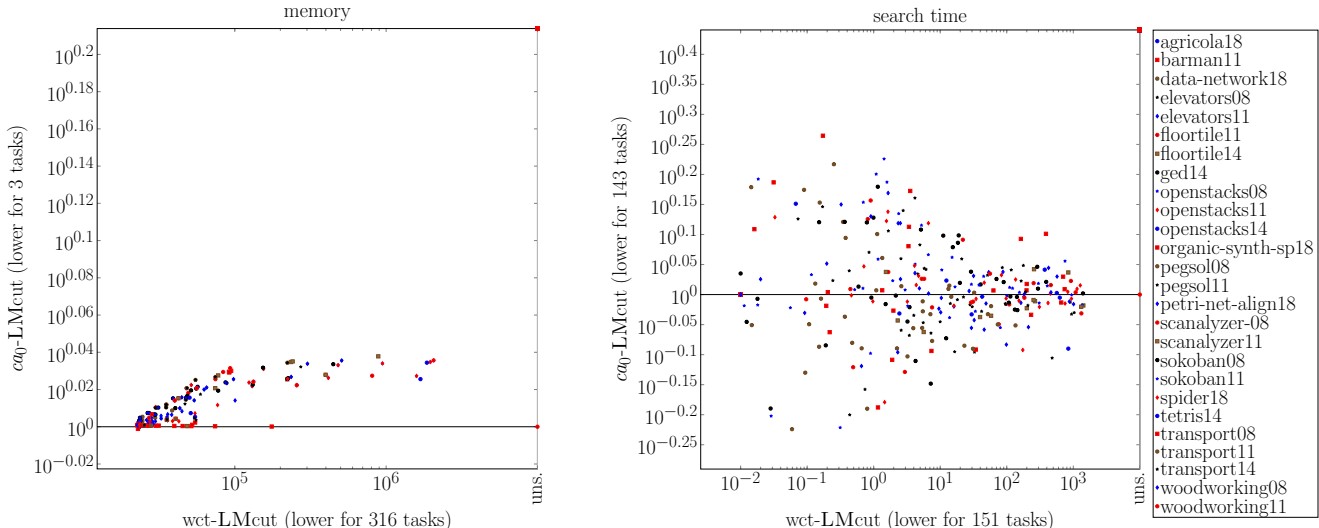

Figure 3: Memory (left) and search time (right), comparing cost algebraic configuration $ca_0$ to cost-transformation configuration *wct*.

| | LMcut | | M&S | | CEGAR | | $h_{max}$ | | iPDB | |
|---|---|---|---|---|---|---|---|---|---|---|
| Coverage | $ca_0$ | wct | $ca_0$ | wct | $ca_0$ | wct | $ca_0$ | wct | $ca_0$ | wct |
| **Sum (587)** | **320** | 319 | 301 | **302** | 293 | **294** | 313 | **317** | 316 | 316 |

Table 3: Comparison $ca_0$ and wct.

3 shows the memory consumption and the search time comparison between the two approaches for the LMcut heuristic, with y-axis depicting performance relative to x-axis. We see that the direct cost-algebraic implementation ($ca_0$) consistently suffers from the memory overhead for representing the pairs for ordering the open list. Time performance varies between $ca_0$ taking approximately $0.6$ and $1.84$ as long as *wct*. Still, Table 3, summarizing the overall coverage of these two approaches, shows that the impact on the overall coverage is low. Both approaches are very much on par, with the only larger coverage difference occurring with the $h_{max}$ heuristic, on pegsol and woodworking domains.

## 6  Conclusions and Future Work

We considered the problem of finding a shortest cost-optimal plan. We explored an algorithm from the literature based on cost algebras and presented a new approach based on a cost transformation. The cost transformation approach requires a suitably chosen parameter $M$. Setting it to a trivial value would lead to very high action costs in the transformed task, potentially causing overflow problems in the planning system. For this reason, we discussed several criteria for alternative lower values that still guarantee to find shortest cost-optimal plans.

For the cost-transformation approach, we can either compute a heuristic in the usual way (based on the cost function of the task) or we can evaluate an admissible heuristic with respect to the original cost function and scale it with factor $M$. We have seen that this yields admissible heuristic estimates for the transformed task. While the native estimates give better guidance, their computation time can be significantly higher than with the scaled heuristic. In the experiments we have seen that it depends on the domain which approach performs better.

For the cost-algebraic approach, the experiments reveal that it usually does not pay off to compute informed estimates for distances but that it is in general better to use the blind distance heuristic instead. In the case of integer action costs and integer heuristic estimates for cost, we can achieve the exact same search behaviour with the cost-transformation approach (Theorem 3). Experimentally, the cost-algebraic variant requires more memory but with no significant impact on coverage.

As this work is the first that is explicitly covering the shortest optimal planning problem, we believe that this is the beginning of a long journey. One interesting avenue for future work is understanding whether one of these methods will excel on a given task and heuristic, without performing any search. Another possible direction is an efficient heuristic computation, either simultaneously computing heuristics for both cost and distance or computing a heuristic for a transformed cost task. Yet another important question is how to determine the cost multiplier M without solving first the cost-optimal variant. It would also be interesting to explore

whether existing search pruning techniques (Pochter, Zohar, and Rosenschein 2011; Domshlak, Katz, and Shleyfman 2012; Alkhazraji et al. 2012) require any adaptation in order to be applied to the cost algebra based A* in a shortest optimal planning setting.

## 7 Acknowledgments

Gabriele Röger has received funding from the European Union's Horizon 2020 research and innovation programme under GA no. 101016442. Moreover, this research was partially supported by TAILOR, a project funded by the EU Horizon 2020 research and innovation programme under grant agreement no. 952215. The paper was improved with the valuable input from the anonymous reviewers at SoCS.

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
