# OpenReview forum: "On Producing Shortest Cost-Optimal Plans"
_icaps-conference.org/ICAPS/2022/Workshop/HSDIP — HSDIP 2022_

### Official Review · Reviewer_wthY · 2022-04-25
**.**

**Confidence:** 4
**Overall Score:** Accept

**Review:**

The paper proposes and analyses a set of techniques for obtaining shortest
among cost-optimal plans.

The paper is clearly written, topically it definitely fits HSDIP, and the
contribution of the paper is certainly interesting. It's a clear accept from
me.

Besides some minor issues (below), I have just few questions/suggestions:

1. Since the paper mentions stubborn sets as a technique using a similar
concept of "strong" optimal plans, I would also suggest looking into
"Operator Mutexes and Symmetries for Simplifying Planning Tasks" by Fiser,
Torralba, Shleyfman, which uses the notion of "strong optimal plans" in
exactly the same meaning as in this paper.

2. I think, there is another option how to look for the shortest cost-optimal
plans that is not discussed in the paper: One could use the cost-algebraic
approach already described in the paper, but transform costs of operators into
pairs by replacing a cost c with <c,1> and define the operation "+" as
<a,b> + <c,d> = <a+b,c+d> and "<=" as <a,b> <= <c,d> iff a < c, or a=c and
b<=d. If I'm not mistaken, all compared heuristics use just + and <= on the
costs. So, replacing costs and all corresponding + and <= operations as
described in the whole planner, setting the g-value of the initial state to
<0,0> should suffice to run FD as a solver for the shortest cost-optimal
planning. In this case, I think, we should get basically the same thing as the
"ct" variant (except it does not depend on M). Or am I mistaken? Do you have
any intuition how would this approach differ from the ones you propose?

3. The weakness of the ct and wct variants is the bound M which we need to
guess somehow. Do you have an idea how to do that automatically (in a
domain-independent manner)? I can think of running the planner twice -- once
for cost-optimal solution and then use its length as M for the transformation
for the second run. Would this approach work? Do you have some insight how to
do this differently/more effectively? There are also some recent interesting
works of Mohammad Abdulaziz from University of Munich on estimating upper
bounds on plan lengths that would be worth discussing (and trying
experimentally -- although I don't know how accessible is the implementation).

4. In the experiments, I assume you checked whether the g/h/f-values are
saturated (I guess to INT_MAX) -- how often did this happen? Or was this an
issue only in the parcprinter domain?

Minor issues:

Proof of Theorem 1: last paragraph has (?) but it should be (*)

The second paragraph under Theorem 1 mixes \varepsilon and \epsilon.

Theorem 3: same-tie breaking => same tie-breaking

---

> ### Author Response · Authors · 2022-04-29
> **Thank you for the questions/suggestions**
>
> Thank you for your comment and suggestions.
>
> 1. Thank you for pointing out the reference.
> 2. If we understand correctly, you are proposing to adapt the existing heuristics to find lower bounds on a new cost function. That would require a theoretical proof of admissibility of the heuristic resulted from the suggested computation adaptation. Assuming that it is indeed admissible, it can be used with our implementation of the cost-algebraic A*. One of the aims of this paper is to facilitate future research in shortest cost-optimal planning, and we are happy to see it happening already.
> 3. It is sufficient to set M to a known length of some optimal plan. In our experiments, we used M=10000, as all known cost-optimal plan lengths for IPC domains are lower than 10000. In general, finding an over-approximation of the cost-optimal plan length is an interesting research question.
> 4. The g/h/f values in our experiments never overflow. However, one of the heuristics, namely cegar, does use hadd heuristic internally. There are 24 cases where the costs of hadd were clamped to 100000000: 20 instances of agricola-opt18-strips (where the approach timed out) and 4 instances of organic-synthesis-split-opt18-strips , two timed out and two solved.

---

### Official Review · Reviewer_qBSj · 2022-04-26
**Relevant topic on cost and length optimality**

**Confidence:** 3
**Overall Score:** Accept

**Review:**

This paper tackles the problem of optimal cost planning while minimizing plan length. The paper proposes two approaches, a variant of A* called cost algebraic A*, using two evaluations and revisiting the definition of reopening, and a transformation of the cost function that ensures standard A* returns cost and length optimal plans. Equivalency results and optimality proofs are included, as well as a detailed experimental evaluation

The paper is well written and discusses a relevant topic that will be of interest in the workshop.

I would only suggest another baseline to understand the impact of caring for length optimality. What would be the plan length difference between the solution found by a standard A* + each heuristic, with respect to ca0 and wct, assuming the same tie-breaking for all approaches? This can further illustrate the optimality-length gap benefit of the proposed approaches with respect to standard optimal planners that care only for cost optimality.

---

> ### Author Response · Authors · 2022-04-29
> **Optimality-length gap**
>
> Thank you for your review. You raise an interesting point.
>
> Different cost-optimal planners might produce cost-optimal plans of different lengths. In fact, some cost-optimal planners might sometimes produce a shortest optimal plan. That does not mean that the optimality-length gap is 0, as they do not *guarantee* to produce such plans. It might be interesting to produce a family of planning tasks that can make a particular approach produce large optimality-length gaps. We will try to come up with such an example. Meanwhile, our experiments do show cost-optimal planners to produce non-shortest cost-optimal plans.